# Hepatic Hedgehog Signaling Participates in the Crosstalk between Liver and Adipose Tissue in Mice by Regulating FGF21

**DOI:** 10.3390/cells11101680

**Published:** 2022-05-18

**Authors:** Fritzi Ott, Christiane Körner, Kim Werner, Martin Gericke, Ines Liebscher, Donald Lobsien, Silvia Radrezza, Andrej Shevchenko, Ute Hofmann, Jürgen Kratzsch, Rolf Gebhardt, Thomas Berg, Madlen Matz-Soja

**Affiliations:** 1Rudolf-Schönheimer Institute for Biochemistry, Faculty of Medicine, Leipzig University, 04103 Leipzig, Germany; fritzi.ott@medizin.uni-leipzig.de (F.O.); christiane.koerner@medizin.uni-leizig.de (C.K.); kimfabienne66@gmail.com (K.W.); ines.liebscher@medizin.uni-leipzig.de (I.L.); ivs.gebh@t-online.de (R.G.); 2Division of Hepatology, Clinic and Polyclinic for Oncology, Gastroenterology, Hepatology, Infectious Diseases, and Pneumology, University Hospital Leipzig, 04103 Leipzig, Germany; thomas.berg@medizin.uni-leipzig.de; 3Institute for Anatomy, Faculty of Medicine, Leipzig University, 04103 Leipzig, Germany; martin.gericke@medizin.uni-leipzig.de; 4Institute for Diagnostic and Interventional Radiology and Neuroradiology, Helios Clinic Erfurt, 99089 Erfurt, Germany; donald.lobsien@helios-gesundheit.de; 5Institute for Neuroradiology, University Hospital Leipzig, 04103 Leipzig, Germany; 6Max Planck Institute of Molecular Cell Biology and Genetics, 01307 Dresden, Germany; radrezza@mpi-cbg.de (S.R.); shevchenko@mpi-cbg.de (A.S.); 7Dr. Margarete Fischer-Bosch Institute of Clinical Pharmacology, University of Tübingen, 70376 Stuttgart, Germany; ute.hofmann@ikp-stuttgart.de; 8Institute of Laboratory Medicine, Clinical Chemistry and Molecular Diagnostics, Faculty of Medicine, Leipzig University, 04103 Leipzig, Germany; juergen.kratzsch@medizin.uni-leipzig.de

**Keywords:** adipose tissue, liver, hepatocytes, crosstalk, Hedgehog signaling, FGF21, metabolically healthy obesity (MHO), sex dimorphism

## Abstract

The Hedgehog signaling pathway regulates many processes during embryogenesis and the homeostasis of adult organs. Recent data suggest that central metabolic processes and signaling cascades in the liver are controlled by the Hedgehog pathway and that changes in hepatic Hedgehog activity also affect peripheral tissues, such as the reproductive organs in females. Here, we show that hepatocyte-specific deletion of the Hedgehog pathway is associated with the dramatic expansion of adipose tissue in mice, the overall phenotype of which does not correspond to the classical outcome of insulin resistance-associated diabetes type 2 obesity. Rather, we show that alterations in the Hedgehog signaling pathway in the liver lead to a metabolic phenotype that is resembling metabolically healthy obesity. Mechanistically, we identified an indirect influence on the hepatic secretion of the fibroblast growth factor 21, which is regulated by a series of signaling cascades that are directly transcriptionally linked to the activity of the Hedgehog transcription factor GLI1. The results of this study impressively show that the metabolic balance of the entire organism is maintained via the activity of morphogenic signaling pathways, such as the Hedgehog cascade. Obviously, several pathways are orchestrated to facilitate liver metabolic status to peripheral organs, such as adipose tissue.

## 1. Introduction

The morphogenic Hedgehog (Hh) pathway is best known for its effects during embryogenesis, where it plays an important role in organ formation and tissue patterning [1]. One of the best-studied tasks in this sense is the sonic hedgehog (SHH) gradient-dependent development of the ventral neural tube in vertebrates [2]. Hh signaling is important for the regeneration of various tissues, such as bone and skin, through the activation of proliferative pathways. However, dysregulated Hh signaling is involved in the formation of tumors, such as basal cell carcinoma and medulloblastoma [3,4], highlighting the importance of balanced signaling pathways. In recent years, it has become apparent that Hh activity is not limited to those developmental processes but also participates in homeostasis and regulation of metabolism in adult organisms. Hh signaling is implicated in the proper function of various vital organs, such as those in the digestive system, the brain, adipose tissue and the liver [5,6,7,8]. In the liver, Hh is an important regulator of energy balance and is deeply involved in the regulation of lipid metabolism [9,10]. Furthermore, Hh is implicated in maintaining structural heterogeneity of the liver via zonation, where other morphogenic pathways, such as Wnt/β-catenin, are also involved [7].

The Hh signaling cascade consists of many proteins whose diverse interactions are nicely summarized in a review by Zhou and Jiang [11]. Very briefly, in vertebrates, three different ligands are known to activate the Hh signaling pathway, Sonic Hedgehog (SHH), Indian Hedgehog (IHH) and Desert Hedgehog (DHH). All three ligands bind to the transmembrane receptor Patched (PTCH 1/2), which reverses its inhibition of the coreceptor Smoothened (SMO). This results in signal transduction through a multiprotein complex consisting of Fused (FU), Suppressor of Fused (SUFU), and the three transcription factors glioma-associated oncogenes 1, 2, 3 (GLI1, 2, 3). These transcription factors ultimately form activator and/or repressor proteins that control target gene expression [12].

Although much important work has been carried out investigating the impact of Hh signaling on homeostasis in adult tissues, relatively little is known about the crosstalk among different tissues that may occur via Hh signaling—or, in other words, how changing Hh activity in one organ affects the homeostasis of another organ. In this regard, our group has already shown that a loss of hepatocellular Hh signaling in female mice leads to a lack of an estrous cycle and dysplasia of the ovaries through the persistence of hepatic steroidogenesis accompanied by androgenization. The resulting infertility is very similar to the clinical outcome of polycystic ovary syndrome [13].

Continuous crosstalk between the liver and adipose tissues is known to ensure stable energy homeostasis; lipids are secreted by the liver for storage in adipocytes in the saturated state, while during fasting they are transported back to the liver [14]. In this regard, hepatokines and cytokines are well-known players that ensure this precise coordination, but the regulatory details are not fully understood. In nonalcoholic fatty liver disease (NAFLD) and nonalcoholic steatohepatitis (NASH), the crosstalk of liver and adipose tissues is well studied. Novel studies have shown that adipose tissue function, rather than adipose tissue expansion, is the main driver of NAFLD in obesity [15].

In this study, we show that modulation of the hepatic Hh pathway by hepatocyte-specific knockout of *Smo* in mice (Hh mice) [9] has an immense effect on white adipose tissue characterized by a strong sex difference. The changes in lipid profiles, hormones and lipoproteins indicate a metabolically healthy obesity (MHO) phenotype [16], which is most likely caused by strong hepatic upregulation of the hepatokine fibroblast growth factor 21 (FGF21). Despite the *Fgf21* gene expression being altered by a wide array of physiological, metabolic, and environmental factors, there were additional changes in several dominant transcriptional regulators, such as the circadian rhythm, the mTOR machinery and genes of the WNT pathway. Using chromatin immunoprecipitation of the Hh transcription factor GLI1, we showed binding sites for many of those genes. Thus, this study provides another piece of the puzzle regarding the importance of the Hh signaling pathway in organ–organ communication.

## 2. Materials and Methods

### 2.1. Maintenance and Feeding of the Mice

Hh (SAC mice) and C57BL/6N mice were housed in a pathogen-free facility under a 12:12 h light–dark cycle according to European (Directive 2010/63/EU) and German guidelines for the care and safe use of experimental animals. All animal experiments were approved by the Landesdirektion Sachsen (permission numbers: TVV11/08, TVV44/16).

The animals were fed ad libitum with regular chow (V1534-300 composed of 24.0 kJ% protein, 67 kJ% carbohydrate, 9 kJ% fat; usable energy: 13.5 kJ/g; ssniff^®^ Spezialdiäten GmbH, Soest, Germany) and tap water throughout life. The mice were euthanized at 12 weeks of age between 9 and 11 am after administration of an anesthetic consisting of ketamine, xylazine and atropine. Only for blood collection for analysis of serum parameters were the mice starved for 24 h followed by returned access to regular chow for 12 h to obtain a synchronized feeding state.

### 2.2. Metabolic Cages

To analyze the physiological parameters of Hh mice, they were housed under controlled temperature and lighting with free access to food and water for four days and three nights. Food/water intake, energy expenditure, respiratory exchange ratio, and physical activity were measured using metabolic cages (TSE Systems).

### 2.3. MRI Analysis

To visualize adipose tissue depots, magnetic resonance imaging (MRI) scans of Hh-WT (wild-type) and Hh-KO (knockout) mice were conducted at University Hospital Leipzig (MR Achieva R3.2, Philips Health care, Hamburg, Germany). Representative scans are shown.

### 2.4. Tissue Isolation

Adipose and liver tissues were isolated from Hh mice. Brown adipose tissue (BAT) was isolated from the interscapular region, gonadal adipose tissue, also called visceral adipose tissue (VAT), from the abdomen and subcutaneous adipose tissue (SAT) from the inguinal region [17].

Tissue was either snap-frozen for RNA isolation or fixed in 4% formaldehyde solution for at least 24 h at 4 °C for histological evaluation.

### 2.5. Isolation and Cell Culture of Primary Hepatocytes

Primary hepatocytes from Hh mice were isolated using a previously described collagenase perfusion method [18]. The cell suspension was cleared of nonparenchymal cells using differential centrifugation [9]. The pure hepatocyte fraction was either snap-frozen for RNA isolation or cultured as described elsewhere [7] for 24 h. The culture medium was collected for FGF21 enzyme-linked immunosorbent assays (ELISAs) and bile acid quantification.

### 2.6. Histological Analysis and Immunohistochemistry

Paraffin-embedded sections (5 µm) of adipose tissue were stained immunohistochemically as described earlier [19]. Samples were permeabilized with three PBS (with 0.3% Triton) washes for 5 min each. Nonspecific binding was blocked by incubating with 1% BSA (with 0.3% Triton) for 1 h at room temperature (RT). The slices were incubated with primary antibody diluted in 1% BSA (with 0.3% Triton) overnight at 4 °C in a humidified chamber. The following primary antibodies were used: rat anti-MAC2 (1:1000, #CL8942AP, Cedarlane, Burlington, ON, Canada) and rabbit anti-perilipin (D1D8) XP (1:200, #9349, Cell Signaling Technologies, Cambridge, UK). For fluorescence staining, an additional incubation with a biotinylated antibody (goat anti-rat 1:1000, #ab6119, Abcam, Cambridge, UK) was implemented in the protocol to enhance the fluorescence signal. The following secondary antibodies were utilized: goat anti-rabbit Alexa Fluor 488 (1:250, #ab150077, Abcam, Cambridge, UK), rabbit anti-chicken (1:250, #303-165-003, Jackson ImmunoResearch, Cambridgeshire, UK) and extravidin antibiotin indocarbocyanine C3 (1:500, #E4142, Merck, Darmstadt, Germany). Biotinylated and secondary antibodies were incubated on slides for 1 h at RT in a humidified chamber. Nuclei were stained using 2.5 µg/mL DAPI diluted in 1x PBS for 20 min at RT. Background fluorescence was minimized using 0.3% Sudan black solution that was heated to 50 °C. After a 2 min incubation step, the sections were washed with distilled water several times and mounted using Mowiol (with 0.1% DABCO). The staining was captured using a Leica DM5000 microscope and LAS 4.12 software (Leica, Wetzlar, Germany).

### 2.7. Adipocyte Size Analysis

Perilipin-1 fluorescence staining was used to visualize the cell membranes of female and male Hh-WT and Hh-KO adipocytes from VAT and SAT. Eight to twelve randomly chosen regions were captured using an Olympus BX51 epifluorescence microscope (Olympus, Hamburg, Germany). Subsequently, the mean diameter of the cells was analyzed using Olympus cellSens Dimension software (Tokyo, Japan).

### 2.8. Shotgun Lipidomics Analysis

Adipose tissue was homogenized in 200 µL IPA (30 min, 4 °C, 30 Hz). The homogenate was centrifuged (5 min, 4 °C, 13,000 rpm), and the supernatant was collected. A blank sample containing only solvent that was spiked with internal standard mixture was generated in parallel. The dried supernatant was dissolved in CHCl_3_/MeOH 2:1 *v/v* (1 mL), and the volume corresponding to 100 µg of wet tissue was removed and dried for further analyses. Lipids were extracted at 4 °C using a two-step chloroform/methanol (first step with 990 µL of chloroform/methanol 17:1, 120 min; second step with 990 µL of chloroform/methanol 2:1 *v/v* for 120 min) extraction adding 200 µL H_2_O to induce phase separation [20,21]. The samples were spiked with an internal lipid standard mixture.

The two-step extracts were dissolved in SprayMix solution (4:2:1 IPA/MeOH/CHCl_3_
*v*/*v*/*v* + 7.5 mM ammonium formate) and analyzed by direct infusion on a Q-Exactive mass spectrometer (Thermo Fisher Scientific Inc., Waltham, MA, USA) equipped with a TriVersa NanoMate ion source (Advion Biosciences, Ithaca, NY, USA) using a nanoelectrospray chip.

Spectra acquired in *t*-SIM mode were preprocessed using repetition rate filtering software PeakStrainer [22] and stitched together using an in-house developed script [23]. Lipids were identified using LipidXplorer software [24]. Molecular fragmentation query language (MFQL) queries were compiled for phosphatidylcholines (PC), ether phosphatidylcholines (PC-O), lyso-phosphatidylcholines (LPC), ether lyso-phosphatidylcholines (LPC-O), phosphatidylethanolamines (PE), ether phosphatidylethanolamines (PE-O), lysophosphatidylethanolamines (LPE), phosphatidylinositols (PI), phosphatidylserines (PS), sphingomyelins (SM), triacylglycerols (TAG), diacylglycerols (DAG) and cholesteryl esters (CE) lipid classes. Lipids were identified by matching accurately determined intact masses (mass tolerance < 5 ppm). Peak intensities were normalized to the intensities of corresponding internal standards. Statistical analyses were performed using the MetaboAnalyst web tool.

### 2.9. Serum Parameters

To reduce the influence of direct food intake on these parameters, we subjected the animals to a 24 h starvation period followed by a 12 h feeding period before blood sampling. Blood was collected from the beating heart of anesthetized mice. Serum samples were analyzed for insulin (#EIA-3439; DRG Diagnostics, Marburg, Germany), leptin (#E06, Mediagnost, Reutlingen, Germany) and adiponectin (#AKMAN-011, Shibayagi Co., Ltd., Shibukawa, Japan) using ELISAs.

Lipoproteins were isolated by sequential ultracentrifugation from 60 µL of plasma at densities (d) of <1.006 g/mL (very low-density lipoprotein, VLDL), d 1.063 g/mL (intermediate density lipoprotein and low-density lipoprotein, LDL) and d > 1.063 g/mL (high-density lipoprotein, HDL) in an LE-80K ultracentrifuge (Beckman Coulter Brea, CA, USA). Cholesterol in the lipoprotein fractions was determined enzymatically using a colorimetric method (Roche, Basel, Switzerland).

### 2.10. Western Blot Analysis

Western blot analysis was performed as previously described [25]. Briefly, proteins were isolated from BAT, SAT and VAT of female and male Hh mice using RIPA lysis buffer and quantified using a BCA assay (Thermo Fisher Scientific Inc., Waltham, MA, USA) according to the manufacturer’s instructions. A total of 5–40 µg of protein was separated using 12% SDS–PAGE (Bio–Rad Laboratories GmbH, Feldkirchen, Germany). The proteins were immunoblotted to a nitrocellulose membrane (LI-COR Biosciences, Lincoln, RI, USA) using a tank blot system (Bio–Rad Laboratories GmbH, Feldkirchen, Germany). The membranes were blocked in Odyssey^®^ blocking buffer in TBS (#927-60001, LI-COR Biosciences) and incubated with the primary antibody rabbit-anti-UCP1 (1:1000, #ab10983 Abcam,) overnight. The secondary antibody IRDye^®^ 800 CW goat anti-rabbit IgG (1:10,000 LI-COR Biosciences, #926-32211) and the Odyssey^®^ Fc imager (LI-COR Biosciences) were used for the detection and quantification of the proteins. Total protein concentrations were determined using REVERT-Total Protein Stain (LI-COR Biosciences), which was used for normalization.

### 2.11. RNA Isolation and Quantitative Real-Time PCR (qPCR)

Total RNA was isolated from cells and tissues using peqGOLD RNAPure™ (VWR International, 30-1010) and innuSOLV RNA Reagent (analytikjena, Jena, Germany) according to the manufacturer’s instructions. A Precellys^®^ 24 (Bertin, Montigny-le-Bretonneux, France) was used (6000 rpm, 30 s) to homogenize the adipose tissue samples. Reverse transcription was performed using a Proto-Script^®^ First Strand cDNA Synthesis Kit (E6300L, New England Biolabs, Ipswich, MA, USA) according to the manufacturer’s instructions. For qPCR analyses, gene-specific and intron-spanning primers were designed using Primer 3 software and were acquired from Microsynth (Balgach, Switzerland) unless otherwise noted (Appendix A). A Biozym Blue S’Green qPCR Kit (331416XL, Biozym, Hessisch Oldendorf, Germany) and a Rotor-Gene Q (Qiagen, Hilden, Germany) were used according to the manufacturer’s instructions. Expression levels were quantified in duplicate and were calculated using internal amplification standards. 14-3-3 Protein zeta/delta (*Ywhaz*) was used as a reference gene for all samples for normalization.

Knockout of *Smo* was confirmed by assessment of *Smo* mRNA in liver tissue or hepatocytes. Only Hh-KO samples with *Smo* expression of less than 50% were included.

### 2.12. cAMP Assay

The AlphaScreen R cyclic adenosine monophosphate (cAMP) Detection Kit (PerkinElmer, #6760635M, Waltham, MA, USA) was used according to the manufacturer’s instructions to analyze tissue samples of BAT, SAT and VAT from Hh-WT and Hh-KO animals. Samples were incubated with 350 µL of LI buffer containing IBMX to inhibit phosphodiesterase activity. Emitted light was measured at wavelengths from 520 nm to 620 nm.

### 2.13. Quantification of Bile Acids

Bile acids were measured using negative electrospray (ESI) LC–MS/MS in multiple-reaction-monitoring (MRM) mode on an Agilent 6460 triple quadrupole mass spectrometer (Agilent, Waldbronn, Germany) coupled to an Agilent 1200 HPLC system. The bile acids tauro-α,β-muricholate, β-muricholate, taurocholate and cholate were separated on a Poroshell 120 EC-C18 column (100 3 2.1 mm, 2.7 mm particle size, Agilent) using a gradient of (A) 12 mm ammonium acetate in water and (B) acetonitrile as mobile phases at a flow rate of 0.5 mL/min. MRM transitions were 514.3/80 for taurocholate and tauro-α,β-muricholate, 407.3/407.3 for β-muricholate and 518.3/80 for the internal standards [2H4]tauromuricholate and [2H4]taurocholate.

### 2.14. FGF21 ELISA

A Mouse FGF21 ELISA Kit PicoKine™ (#EK1379, Boster Biological Technology, Pleasanton, CA, USA) was used according to the manufacturer’s instructions. One hundred microliters of hepatocyte supernatant or serum from Hh mice was used. The absorbance of the samples was measured at 450 nm using a GloMax^®^ Discover Microplate Reader by Promega (Madison, WI, USA). Data were analyzed using a standard curve.

### 2.15. Chromatin Immunoprecipitation with Sequencing (ChIP-Seq)

Hepatocytes from four male C57BL/6N mice were pooled and used for ChIP-seq performed by the company Active Motif (Carlsbad, CA, USA). In brief, hepatocytes were fixed in 1% formaldehyde for 15 min and quenched with 0.125 M glycine. Chromatin was isolated by adding lysis buffer, followed by disruption using a Dounce homogenizer. The lysates were sonicated, and the DNA was sheared to an average length of 300–500 bp using Active Motif’s EpiShear probe sonicator (#53051). Genomic DNA (input) was prepared by treating aliquots of chromatin with RNase and proteinase K and heating for decrosslinking, followed by SPRI bead clean up (Beckman Coulter, Brea, CA, USA) and quantification by Clariostar (BMG Labtech, Ortenberg, Germany). Extrapolation to the original chromatin volume allowed determination of the total chromatin yield.

An aliquot of chromatin (40 µg) was precleared using protein A agarose beads (Invitrogen, Waltham, MA, USA). Genomic DNA regions of interest were isolated using 6 µg of antibody against Gli1 (#GTX124274, GeneTex, Irvine, CA, USA). The complexes were washed, eluted from the beads using SDS buffer, and subjected to RNase and proteinase K treatment. Crosslinks were reversed by incubation overnight at 65 °C, and ChIP DNA was purified by phenol–chloroform extraction and ethanol precipitation.

Illumina sequencing libraries were prepared from the ChIP and input DNAs on an automated system (Apollo 342, Wafergen Biosystems/Takara, Fremont, CA, USA). After a final PCR amplification step, the resulting DNA libraries were quantified and sequenced on an Illumina NextSeq 500 (75 nt reads, single end). Reads were aligned to the mouse genome (mm10) using the BWA algorithm (default settings). Duplicate reads were removed, and only uniquely mapped reads (mapping quality ≥ 25) were used for further analysis. Alignments were extended in silico at their 3′-ends to a length of 200 bp and assigned to 32-nt bins along the genome. Peak locations were determined using the MACS algorithm (v2.1.0) with a cutoff *p*-value = 1 × 10^−7^ [26]. Peaks that were on the ENCODE blacklist of known false ChIP-Seq peaks were removed.

### 2.16. Statistics

Statistical analyses and chart design were performed using GraphPad Prism 9. Outliers were identified using the ROUT method with aggressiveness set to 1%. The cleaned data were used for subsequent statistical analyses and data depiction.

Data are plotted as the mean values of biological replicates ± standard deviation. For statistical analysis of experiments (multiple), unpaired *t*-tests were performed. The null hypothesis was rejected at (corrected) *p*-values of * *p* < 0.05, ** *p* < 0.01, *** *p* < 0.001 and **** *p* < 0.0001.

## 3. Results

### 3.1. Hh Knockout in Hepatocytes Is Accompanied by Phenotypic Changes in Adipose Tissue

The transgenic mouse line with hepatocyte-specific deletion of the *Smo* gene (abbreviated Hh-KO mice) was previously described in detail [9]. As already published, Hh-KO mice were generally smaller, and their body weight was significantly reduced in both male and female Hh-KO mice compared to Hh-WT mice [9]. To our surprise, MRI scans of Hh mice revealed an increase in adipose tissue in female and male knockout animals, represented by white areas in the MRI scans (Figure 1A). To validate this observation, we isolated the tissue types brown adipose tissue (BAT), subcutaneous adipose tissue (SAT) and visceral adipose tissue (VAT) and analyzed their weights. The data revealed that all subtypes of adipose tissue were significantly increased in Hh-KO mice of both sexes compared to Hh-WT mice in relation to body weight (Figure 1B).

To verify whether the increased adipose tissue resulted from physiological changes, the animals were observed in metabolic cages for three nights and four days to monitor food intake, energetic expenditure and physical activity (Appendix A). Food intake (Appendix A), the respiratory exchange ratio (Appendix A) and physical activity represented as running distance (Appendix A) exhibited no differences between the genotypes. Only the Hh-KO females displayed slightly lower energy expenditure during the day than the Hh-WTs, while no changes were found in males (Appendix A).

The white adipose tissues were further characterized using perilipin staining to investigate the adipocyte diameter (Figure 1C and Appendix A), which exhibited a significant increase in size in SAT and VAT from Hh-KO mice in both sexes compared to WT (Figure 1C). In detail, we observed a significant shift of relative frequency from the lower range of cell sizes to adipocytes >70 µm (Appendix A) in male and female SAT from Hh-KO mice. The same observation was made for adipocyte cell size in VAT, although only in females was the increase in the relative frequency of larger cells significant (Appendix A).

Since the enlargement of adipose tissue is often accompanied by inflammatory processes [27], we investigated macrophage accumulation via MAC2 staining. However, a significant raise was not detected for increased infiltration with macrophages in SAT or VAT (Appendix A).

### 3.2. Lipidomic Signature Reveals Fundamental Sex Differences Regarding Adipose Tissue Composition Due to Hepatic Hh Deletion

For a detailed investigation of the changes in adipose tissue via hepatocyte-specific deletion of *Smo*, we employed a shotgun lipidomic analysis of SAT and VAT. The analysis reported a molar abundance of 287 species of 12 lipid classes (TAG, DAG, CE, SM, PE, PE-O, PC, PC-O, LPE, LPC, PI and PS), including 287 lipid species. First, partial least squares-discriminant analysis (PLS-DA) was used to cluster Hh-WT and Hh-KO mice using the normalized intensity peak data. The PLS-DA revealed strong discrimination between the Hh-WT and Hh-KO, whereby the variable importance in projection (VIP) score indicated different lipids as the most contributory variables in class discrimination (Appendix A).

In line with previous results, we observed sex-specific alterations with more distinct changes in the SAT and VAT from female Hh-KO mice than in those of Hh-WT and male mice. Interestingly, we found no significant changes in TAG concentration in SAT or VAT in either sex (Figure 2), but its precursor molecule DAG was significantly less abundant in SAT from female Hh-KO mice (Figure 2A). Additionally, the levels of CEs, SM, LPE and PS were significantly reduced in SAT from Hh-KO female mice. In contrast, the abundance of PE and PE-O in SAT in females was significantly increased (Figure 2A). With respect to the levels of lipid species, however, a more differentiated picture emerged. Interestingly, the most altered lipid class in female SAT included multiple TAG species, such as TAG40:1, TAG 36:0 and TAG 38:1 (Appendix A; Appendix A). Furthermore, the heatmap shows that the top 25 lipid species that were significantly downregulated due to hepatic Hh deletion included membrane molecules, such as PC, PE and SM (Figure 2A).

When looking at the differences between Hh-KO and WT in male SAT, there were no significant changes (Figure 2B). However, we observed only a few significantly increased lipids among TAG species, such as TAG 45:5, TAG 46:5 and TAG 44:4, whereby it is worth noting that their regulation was completely reversed compared to the data in female SAT (Figure 2B and Appendix A; Appendix A).

We also observed these strong sex-specific differences in VAT from Hh mice. Therefore, we revealed that in total, female Hh-KO mice exhibited significantly reduced levels of PC-O, LPE and PI, while no significant changes were observed in the males (Figure 2C,D). Furthermore, the heatmap of the 25 top lipid species again included significantly reduced levels of cell membrane components, such as LPC, LPE, PE and PC, in female VAT (Figure 2C). In contrast, the few upregulated lipid species included almost exclusively DAG species (Figure 2C and Appendix A; Appendix A).

In males, the abundance of only five lipid species were significantly changed, which are all components of the membrane lipid spectra, such as PS (36:1, 40:4, 38:5) and PG (34:1, 40:7) (Appendix A; Appendix A). The heatmap for the VAT from the males is very similar to the pattern in male SAT. Here, too, the influence of the deletion of the Hh signaling pathway in hepatocytes seems to be much less pronounced than in females (Figure 2D).

### 3.3. Hormonal and Lipoprotein Changes Indicate Metabolically Healthy Obesity (MHO) in Hh-KO Mice

Since the results of the lipidomic analyses of Hh-KO mice revealed a rather atypical outcome in the case of severe obesity, as suggested by the analysis of MRI and relative fat mass (Figure 1A,B), we performed further investigations at the hormonal level. We initially focused on the hormones insulin, leptin and adiponectin, as these are known to be directly associated with obesity and are often linked to a type 2 diabetes (T2D)-like phenotype. Interestingly, the levels of insulin were significantly decreased in both female and male Hh-KO animals compared to WT animals, whereas adiponectin and leptin were significantly elevated in both sexes of Hh-KO mice (Figure 3A,B). These findings are contrary to the definition of metabolically unhealthy obesity (MUO) and the known changes in T2D, therefore, it is reasonable to assume that the mice developed a metabolically healthy obesity (MHO) phenotype [28]. This initial assumption was confirmed by analyzing the lipoproteins HDL, LDL, VLDL and total cholesterol in the serum of Hh mice. In detail, we observed a significant increase in HDL cholesterol (HDL-C) and significantly reduced HDL-triacylglycerols (HDL-TAG) in female Hh-KOs, whereas no changes were detectable in males (Figure 3C,F). With respect to LDL-C and LDL-TAG levels, we only found a significant reduction in the male animals. No changes were observed in the levels of VLDL-C, VLDL-TAG or total cholesterol (Figure 3C,F).

### 3.4. Gene Expression Profile Changes Indicate Strong Upregulation of Lipid Synthesis and Thermogenesis in VAT from Female SC-KO Mice Due to Hepatocyte-Specific Hh Deletion

A frequently observed phenomenon with respect to changes in adipose tissue is the appearance of a large increase in the number of brown adipocyte-like cells termed beige or brite (brown-in-white) adipocytes in white adipose tissue WAT, which is associated with increased thermogenesis. This effect, called “browning”, is accompanied by strongly upregulated expression of multiple transcription factors, such as the PR domain containing 16 (PRDM16) and the uncoupling protein (UCP)-1 [29]. To determine whether the deletion of hepatic Hh signaling leads to such an effect, we focused on gene expression signatures specific for brown, white and beige adipocytes. Furthermore, we investigated genes that encode central enzymes in lipid synthesis as well as essential lipogenic transcription factors and regulators.

The results showed that inhibition of hepatocyte-specific Hh signaling led to the significantly stronger expression of the *Ucp1* and *Cox8b* (cytochrome c oxidase subunit 8B) genes in female SAT and VAT, whereas male Hh-KO animals exhibited no significant change in these genes. Only in the BAT of male Hh-KOs was *Cox8b* significantly decreased. The transcription factor *Ppara* (peroxisome proliferator-activated receptor alpha) was significantly upregulated in BAT but not in SAT or VAT of both sexes. Interestingly, *Mpzl2* (myelin protein zero-like 2, also known as *Eva1*), which is a typical gene for murine BAT, was significantly reduced in BAT and SAT in male mice, whereas in female Hh-KO mice, it was not changed. Another marker gene for browning, *Tmem26* (transmembrane protein 26), was significantly decreased in females and increased in males (Figure 4A).

Genes encoding proteins essential for lipid synthesis (e.g., *Fasn* (fatty acid synthase), *Agpat* (1-acylglycerol-3-phosphate O-acyltransferase 1) and *Pdk4* (pyruvate dehydrogenase kinase, isoenzyme 4)) exhibited a partially significant increase in female SAT, whereas no change was detected in males, with the exception of *Fasn,* which was surprisingly significantly decreased (Figure 4B). With the exception of *Agpat* in males, we did not observe any significant changes in the expression of the investigated genes in VAT of both sexes.

Regarding essential transcription factors for lipid synthesis and adipogenesis, we detected an increase in *Srebf1a* (sterol regulatory element-binding transcription factor 1), *Srebf1c* and *Pparg* in female SAT, whereas a decrease in all these genes was observed in male Hh-KO mice (Figure 4B).

Since increased thermogenesis and lipid synthesis were observed in the SAT of female Hh-KO animals at the gene expression level, we assessed whether this change in *Ucp1* also occurred at the protein level to lead to heat production in white adipose tissue. For this reason, we performed a Western blot analysis and measured the levels of cyclic adenosine monophosphate (cAMP) in the adipose tissue of the animals. The results showed that there was only a slight increase in UCP1 in the SAT of the female Hh-KO mice and that there was no difference in cAMP in any of the adipose tissue species examined in males or females (Appendix A).

To determine Hh activity in adipose tissue, central genes of the pathway were quantified using qPCR. When examining differences between the sexes, it became apparent that especially in the BAT genes, such as *Shh*, *Disp2*, *Fu*, *Sufu* and *Gli1*/*2/3,* female Hh-KO animals exhibited an opposite regulatory trend compared to male Hh-KOs (Appendix A). Furthermore, it was evident that almost all genes of Hh signaling investigated in VAT displayed a tendency for upregulation in male Hh-KO mice, whereas only a few genes were slightly increased in female KO animals. We found the greatest difference in expression of the receptor *Ptch1* in SAT from female Hh-KO mice, which was significantly upregulated (Appendix A).

### 3.5. Loss of Hepatic Hh Orchestrates FGF21 Secretion via Its Transcription Factor GLI1

Since the changes described thus far in adipose tissue result from the hepatocyte-specific KO of Hh signaling, it could be concluded that a secreted molecule may enable crosstalk between both tissues. In addition to the potential candidates already investigated, such as adiponectin and leptin, bile acids and FGF21 could be promising candidates [17].

Therefore, we analyzed bile acids in both the serum of Hh-KO and WT mice and medium supernatant from hepatocyte cell cultures. Significant changes were not detected in serum or culture supernatants for the bile acids tauro-α,β-muricholate, β-muricholate, taurocholate and cholate (Appendix A).

In contrast, concentrations of FGF21 in the serum were two-fold higher in male Hh-KO mice than in WT mice (Figure 5A). The difference became even more dramatic in the supernatant of cell cultures from Hh mice, where FGF21 levels were three times higher in the supernatants of Hh-KO compared to Hh-WT animals (Figure 5B).

Since the Hh signaling pathway largely regulates the expression of genes by changing transcription factors, we performed a ChIP-seq analysis of the transcriptional activator GLI1, which is significantly downregulated in Hh-KO mice [9]. The analysis was performed in male C57BL/6N mice to identify a principal mechanistic link between GLI1 and *Fgf21* expression. Even though no direct binding of the *Fgf21* gene by GLI1 was detected, an indirect regulatory effect due to up- or downstream-located response elements cannot be excluded, as ChIP-seq identified binding sites for GLI1 in various genes known to regulate *Fgf21* expression. In detail, we found direct binding sites for GLI1 upstream, downstream or directly in the gene for central genes in the mTOR machinery (*Mtor, Rictor*), circadian rhythm (*Arntl, Per1, Cry, Nr1d2*) and Wnt signaling (*Sufu, Wnt10b, Lrp, Apc, Ctnnbl1, Fzd6, Cdk2/12*) (Figure 5C; Appendix A). The plethora of binding sites found clearly shows that morphogenic pathways, such as Hh signaling, control a network of metabolically relevant cascades via the regulation of its transcription factors, which are ultimately responsible for the maintenance of homeostasis and the communication of the liver with peripheral tissues.

## 4. Discussion

While it is generally accepted that the liver communicates with adipose tissue to maintain energetic homeostasis, the role of hepatic morphogens such as the Hh cascade is still unknown. Our primary findings demonstrate a new manner of crosstalk between the liver and adipose tissues via hepatic Hh signaling and its targeted FGF21 secretion to promote metabolic adaptations of SAT and VAT.

In addition to the known function of the Hh signaling in the liver during regeneration processes and chronic diseases [30], the Hh pathway is involved in hepatic metabolism as well. On the one hand, central genes of the Hh pathway are upregulated as a result of the administration of high-fat diets [31,32]. On the other hand, direct activation of the Hh signaling pathway leads to strong inhibition of hepatic lipogenesis [33]. In line with these results, our group generated transgenic mouse models—such as the Hh-mice used in this study—to demonstrate that deletion of Hh signaling in hepatocytes results in an upregulation of lipogenic transcription factors and enzymes in the liver, which ultimately leads to hepatic lipid overload and finally NAFLD [7,9,10,25]. Interestingly, despite a significantly reduced body weight compared to the wild-type mice [9], these animals exhibited a large increase in brown, subcutaneous and visceral adipose tissue with no reduction in behavioral activity (Figure 1 and Figure 2). Furthermore, the changes we observed in the adipose tissue in knockout mice do not correspond to the typical changes observed in adipose tissue in obese patients, where NAFLD is a consequence of obesity associated with specific changes in gene expression and hormone levels in the lipidomic profile as well as inflammatory processes in the adipose tissue and a T2D phenotype.

Inflammatory processes in adipose tissues are the result of several intrinsic signals caused by obesity-induced tissue expansion and are known to be the link between obesity and insulin resistance in T2D in mice and humans [27]. Since macrophage accumulation is associated with obesity-induced inflammation [34,35], we used this marker, but no significant increase in macrophages in the adipose tissue was detected in SAT or VAT (Appendix A).

Additionally, the changes in lipid composition in Hh-KO mice did not correspond to those found in human individuals with T2D-associated obesity, although we observed a more distinct phenotype between male and female mice. Grzybek et al. reported that obesity in mice fed a high-fat diet leads to an increase in TAGs containing 54–56 carbon atoms, while the levels of TAGs with 48–50 carbon atoms significantly decrease. In our female animals, we observed a decrease in TAGs in the range of 48–50 carbon atoms but an increase only in TAGs 48:4 in the SAT. In male mice, more TAG species were increased, whereas no significant decrease was observed. A similar correlation is known for DAG levels, where the intracellular increase in DAG is associated with altered insulin sensitivity, as this positively correlates with impaired insulin signaling [36]. Since in both female and male Hh-KO mice the total levels of DAG in the SAT were reduced with no significant changes in the VAT observed, we also observed evidence that no metabolic obesity with T2D phenotype developed due to deletion of the Hh signaling pathway in the liver. Additionally, the reduced level of SM in Hh-KO mice, which were more pronounced in females, are in line with findings in the obese *ob/ob* mouse model, which is characterized by massive obesity caused by a mutation in the leptin gene which is the main regulator for food intake [37]. Via the complete loss of leptin the *ob/ob* mice develop a mild insulin resistance compared to other mouse models [38] and corresponds to the observation that a nondiabetic phenotype is associated with a reduced level of SM in SAT in human obese patients [39].

Closely related to changes in the lipid composition of adipose tissues is the modification of their metabolic behavior. To investigate this, we analyzed UCP1 and cAMP. UCP1 promotes uncoupled mitochondrial respiration to substantially increase energy expenditure, accompanied by cAMP production, to sustain body temperature in adipose tissue. Although UCP1 is found in very large quantities in BAT, certain circumstances can lead to the presence of brown fat-like adipocytes in white adipose tissue, which is called browning or beiging. Even though cold exposure, β3-selective adrenergic agonists and exercise are the primary triggers for browning, recent work has shown that beige adipocytes also occur naturally in white adipose tissue and are more likely to be responsible for maintaining metabolic status [40,41]. Thereby, the accumulation of beige adipocytes in white adipose tissue leads to an increase in overall energy expenditure and substrate metabolism, increases glucose tolerance and thus contributes to a healthier metabolic phenotype [42]. Despite the fact that we observed a significant increase in *Ucp1* gene expression in SAT and VAT of the female Hh-KO mice, we did not observe a significant increase in UCP1 protein or increased cAMP levels in any of the adipose tissues (Appendix A), indicating that no beiging occurs due to Hh deletion in hepatocytes. This interpretation is supported by the ambiguous expression of other genes that play an important role in beiging, such as *Prdm16*, *Cox8b*, *Tmem26* and *Ppara* [41]. In addition, the results from gene expression to identify factors playing an important role in de novo lipogenesis (DNL) indicate a tendency of a principal downregulation in BAT, SAT and VAT, which was more pronounced in male Hh-KO mice. This downregulation in the DNL was observed in several studies and could be a late adaptive process aimed at limiting the further accumulation of fat mass [43,44,45]. With respect to females, the increased expression of many of the genes examined, especially in SAT, could indicate that there is still capacity for adipose tissue expansion.

To investigate the effects of the observed changes in adipose tissue on a more global level, we measured relevant hormones such as adiponectin, leptin and insulin in the serum of the mice. Interestingly, we did not find increased levels of insulin or decreased levels of leptin or adiponectin, which are hallmarks of T2D-associated obesity. Instead, the reduced levels of insulin and increased serum levels of leptin and adiponectin found in both females and males (Figure 3), combined with the changes in the adipose tissue described above, suggest a phenotype known as “metabolically healthy obesity” (MHO) [16]. The concept of MHO describes an expansion of adipose tissue that is the result of an increased storage capacity, attenuated inflammation of the adipose tissue, elevated secretion of adiponectin and leptin and dyslipidemia [46,47]. In addition, the unchanged bile acid concentrations in Hh mice are a further indication of the MHO phenotype, as in MUO, the intestinal microbiome is markedly altered, which would accompany changes in bile acids [48].

Although the results presented here are largely consistent with the MHO phenotype, there are some differences from the clinical outcomes observed in humans, which can also be observed in other transgenic animal models of MHO [49]. First, we observed hypertrophy of adipose tissue in SAT and VAT in our Hh-KO mice, which is not typical of the MHO phenotype. The expansion in a hyperplastic manner that is observed in MHO is accompanied by a higher proportion of relatively small adipocytes in the adipose tissue [50,51]. Second, the changes in lipoprotein profile only fit in terms of the observed LDL-TAG decrease in Hh-KO mice, which was also observed in humans [52]. However, the increased HDL-C and decreased HDL-TAG levels are contradictory to the MHO phenotype and do not fit the overall association between obesity, which leads to decreased HDL-C levels [53]. However, an increase in HDL-C is known to be cardioprotective, which results in a healthier obese phenotype in our Hh-KO mice [53].

In addition, our findings are all in line with recently published work from Liu et al., which nicely demonstrated that genetically driven NAFLD is a causal risk factor for T2D but with a unique phenotype characterized as normal insulin sensitivity that is further associated with reduced insulin secretion, thus likely representing a late-onset type I-like diabetic phenotype. Genetically induced NAFLD also protects against overall or generalized obesity (as indexed by BMI) but increases the risk for central obesity [54]. Since deletion of the Hh signaling pathway in this model takes place specifically in hepatocytes and a tendency toward the increased activity of the signaling pathway is seen in the investigated adipose tissue species, we assume that the strong increase in adipose tissue is more a consequence of the Hh deletion in the liver rather than a direct consequence of NAFLD in Hh-KO mice. At this point, it should be noted that molecules of the Hh signaling pathway can be considered biomarkers for obesity, such as the Hedgehog-interacting protein (HHIP1), which is significantly reduced in the serum of obese patients [55]. However, the source of HHIP1 is the adipose tissue itself as it is secreted as a negative feedback molecule by the adipocytes to prevent further adipocyte differentiation [56]. The underlying mechanism of our observations appears to be the influence of the Hh pathway on the expression of FGF21, which is already known as a signaling molecule secreted by the liver that communicates with peripheral organs [57,58]. Both in vivo and in vitro, we demonstrated that deletion of the Hh pathway in the liver results in increased hepatocellular secretion of FGF21, which is ultimately reflected by significantly increased FGF21 serum levels in Hh-KO mice. This increased expression of FGF21 causes all of the changes we found in our mice that are reflected in the MHO phenotype [59]. Although we did not observe a direct binding site between FGF21 and the Hh transcription factor GLI1 at the transcriptional level, we found a number of binding sites of the Hh target gene activator GLI1 in promoter regions that regulate the transcription of FGF21. One good example is the circadian clock machinery, which is strongly coupled to the rhythmic expression of FGF21 [60]. Since GLI1 has binding sites in three central genes of the circadian rhythm and deletion of the Hh pathway leads to a significant downregulation of GLI1, we know from previous results that alterations in the Hh pathway have a dramatic effect on the liver’s circadian rhythm [61]. The same conclusion can be made when considering the binding sites of GLI1 in genes of the mTOR and WNT pathways. Using completely independent mouse models as well as several in vitro experiments, our group was able to demonstrate the influence of Hh signaling on these important signaling cascades in recent years [7,25]. Both the mTOR [62,63] and Wnt [64] pathways are known to directly influence FGF21 secretion in hepatocytes. Another regulatory element for the hepatic expression of FGF21 is the transcription factor PPARA [65]. Since these transcription factors are also significantly increased in the hepatocytes of Hh-KO mice [9], they certainly contribute to the increased FGF21 levels. A direct influence of the other Hh transcription factors GLI2 and GLI3 on FGF21 expression is also quite possible. Since those function as activator or repressor proteins, depending on splicing, an analysis of binding sites requires more complex experiments that are currently under investigation in our lab.

The positive effects of FGF21 have been known for over a decade, as administration in obese ob/ob mice leads to weight loss and improvement of hepatosteatosis and metabolic markers [66]. Novel FGF21 analogs are being studied as potential candidates in the treatment of obesity and liver-associated disease such as NASH [67]. Clinical trials with human subjects are underway and generate promising results. Patients suffering from T2D show improvement in markers of insulin sensitivity such as insulin and C-peptide upon administration of the FGF21 analog AKR-001 and positive trends regarding the lipoprotein profile including TAG [68]. The PEGylated human FGF21 analog pegbelfermin, which underwent phase 2 studies, improved metabolic parameters in patients with NASH. It led to increased adiponectin levels similar to observations made in Hh-KO mice and reduced TAG levels [69]. The long-acting FGF21 analog PF-05231023 caused a significant decrease in body weight as well as an increase in adiponectin levels. It also improved lipoprotein profiles in obese patients suffering from T2D [70]. All these results suggest suitability for treatment of metabolic syndrome-related disorders with FGF21 analogs, which is in accordance with our results in mice that show an MHO phenotype due to increased FGF21 expression.

Taken together, these findings demonstrate that the regulation of FGF21 by one or a combination of these factors could be the mechanism through which hepatic Hh signaling ultimately influences adipose tissue and the development of the MHO phenotype in Hh-KO mice.

## Figures and Tables

**Figure 1 cells-11-01680-f001:**
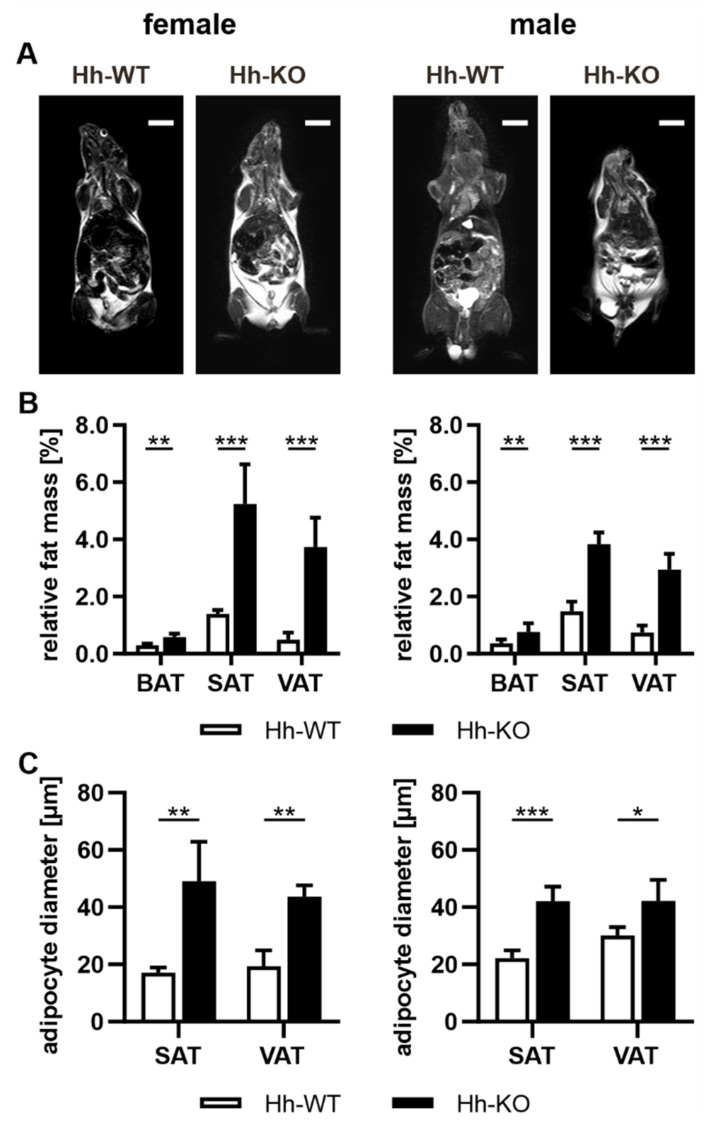
Phenotypic changes in adipose tissue in Hh mice. (**A**) MRI scans showing adipose tissue depots as white areas in female and male Hh-WT and Hh-KO mice. Scale bars indicate 10 mm. (**B**) Quantitative analysis of brown (BAT), subcutaneous (SAT) and visceral (VAT) adipose tissue plotted as a percentage of body weight in female and male Hh-WT and Hh-KO mice; *n* = 4–7. (**C**) Analysis of adipocyte diameter based on perilipin staining (more details in Appendix A) in SAT and VAT from female and male Hh-WT and Hh-KO mice; *n* = 3–4. Multiple unpaired t-tests with *p*-values * ≤ 0.05, ** ≤ 0.01, *** ≤ 0.001.

**Figure 2 cells-11-01680-f002:**
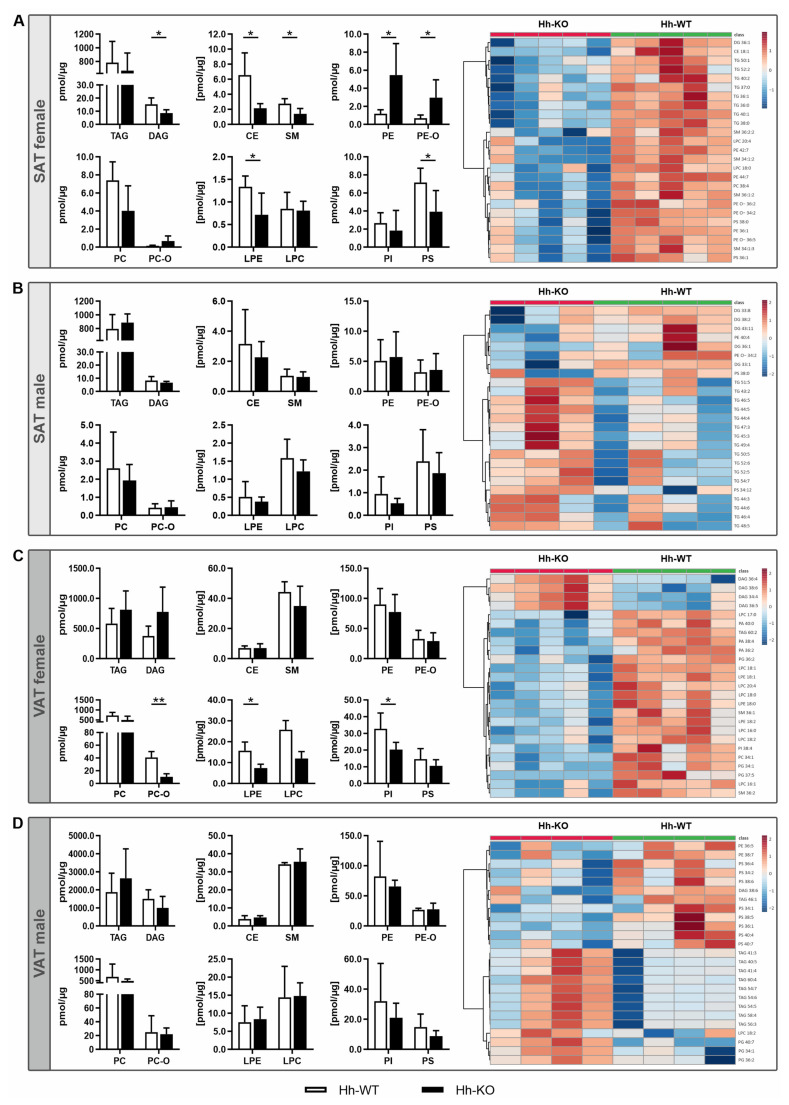
Lipidomic profiles of adipose tissue. Shotgun lipidomic analyses were performed on SAT and VAT from male and female Hh-WT and Hh-KO mice. *n* = 3–5. (**A**–**D**) Quantification of triacylglyceride (TAG), diacylglyceride (DAG), cholesteryl esters (CE), 1,2-diacyl-sn-glycero-3-phosphocholine (PC), 1,2-diacyl-sn-glycero-3-phosphoethanolamine (PE), 1-alkyl-2-acyl-sn-glycero-3-phosphoethanolamine (PE-O), 1-alkyl-2-acyl-sn-glycero-3-phosphocholine (PC-O), 1,2-diacyl-sn-glycero-3-phospho-(1’-myo-inositol) (PI), 1-acyl-sn-glycero-3-phosphocholine (LPC), 1-acyl-sn-glycero-3-phosphoethanolamine (LPE), sphingomyelin (SM), and 1,2-diacyl-sn-glycero-3-phosphoserine (PS) in pmol/µg protein and heatmaps of the top 25 most abundantly regulated lipid classes. *p*-values * ≤ 0.05, ** ≤ 0.01.

**Figure 3 cells-11-01680-f003:**
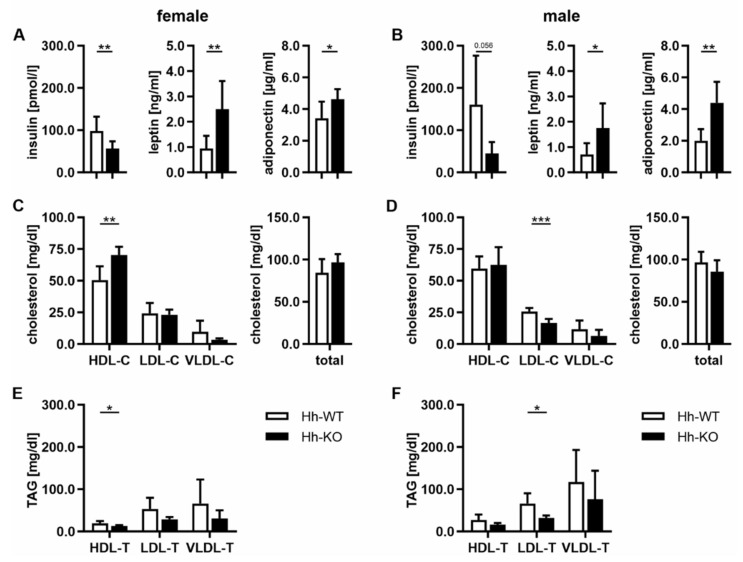
Serum changes in hormone and lipoprotein levels. Serum concentrations of (**A**,**B**) insulin, leptin and adiponectin; (**C**,**D**) cholesterol (C) and total cholesterol and (**E**,**F**) triacylglycerols (T) fractions in HDL, LDL, VLDL were quantified in female and male Hh-WT and Hh-KO mice. *n* = 5–15, (multiple) unpaired t-tests with *p*-values * ≤ 0.05, ** ≤ 0.01, *** ≤ 0.001.

**Figure 4 cells-11-01680-f004:**
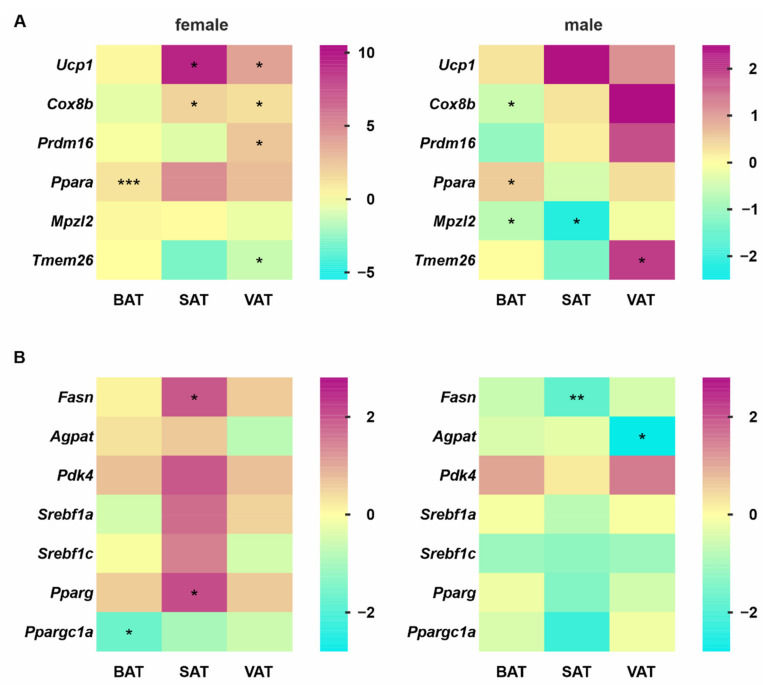
Lipid metabolism-related gene expression profiles. (**A**) Expression of browning and (**B**) lipid synthesis-related genes in female and male Hh adipose tissue was quantified by qPCR. Values are plotted as log_2_ fold changes, *n* = 4–6, multiple unpaired t-tests with *p*-values * ≤ 0.05, ** ≤ 0.01, *** ≤ 0.001. Numeric values are available in Appendix A.

**Figure 5 cells-11-01680-f005:**
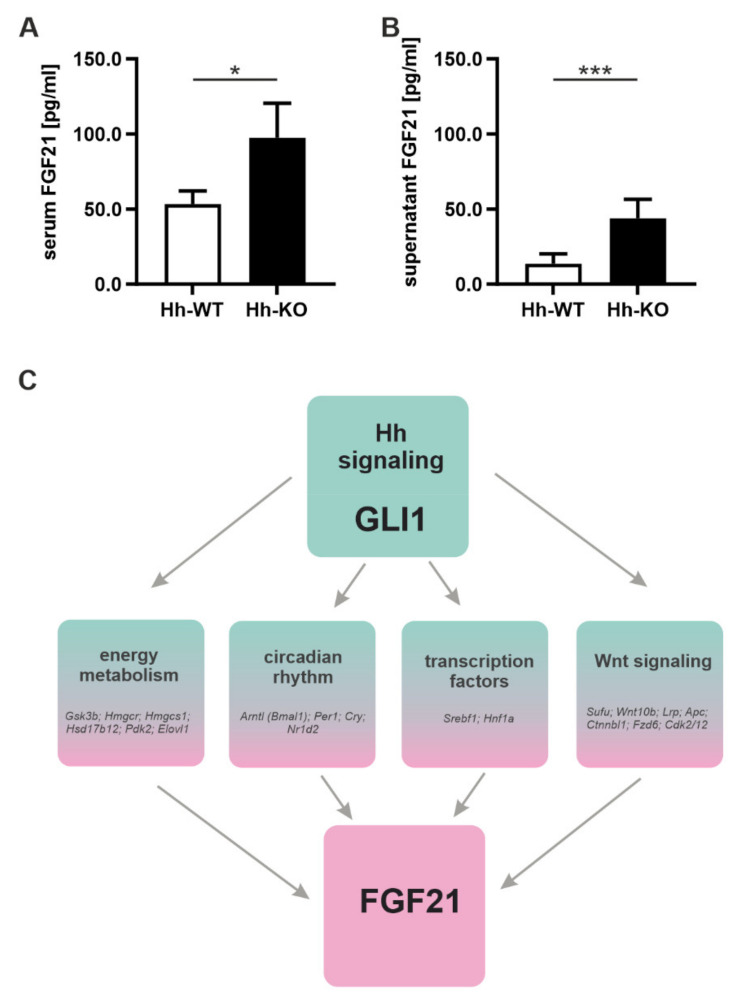
Secretion of FGF21 by hepatocytes. FGF21 was quantified in (**A**) serum and (**B**) hepatocyte supernatants from male Hh-WT and Hh-KO mice by ELISA. Data are plotted as the mean ± standard deviation, *n* = 3–8, unpaired t-tests with *p*-values * ≤ 0.05 and *** ≤ 0.001. (**C**) Binding sites of GLI1 in potential regulators of *Fgf21* expression identified by ChIP-seq. Numerical values are available in Appendix A.

## Data Availability

Not applicable.

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
