# Peer review of "Hepatic Hedgehog Signaling Participates in the Crosstalk between Liver and Adipose Tissue in Mice by Regulating FGF21"

_cells, 2022, doi:10.3390/cells11101680_

Round 1

Reviewer 1 Report

Dear Editor

Thank you very much for giviming me the opportunity to review the paper.

The original research submitted to your journal has a merit to discuss a really intriguing issue in NASH,NALF and liver disease metabolic related. They have a solid scientific basis and a storng laboratory findings

I would just suggest to better discuss in conclusion their results ccording to feww litterature evidences on relation between morphogenic Hedgehog and NAFLD

Author Response

We thank all reviewers for their time to evaluate our manuscript and the useful advices for improvement! Please find our reply below.

Reviewer 1:

  • I would just suggest to better discuss in conclusion their results ccording to feww litterature evidences on relation between morphogenic Hedgehog and NAFLD.

Reply:

We thank the reviewer for this important recommendation. We have expanded the discussion section to include more literature on the subject (line 503-506).

Reviewer 2 Report

This manuscript studies an important topic in the field of hepatology. Few considerations should be highlighted:

There should be more clinical application of the results found here.

Regarding the conclusion; it should be in a separate paragraph or section demonstrating a more clear "clinical" application of the results found.

There should be a section for the abbreviations.

No number for the reference in Line 588.   

Author Response

We thank all reviewers for their time to evaluate our manuscript and the useful advices for improvement! Please find our reply below.

Reviewer 2:

  • Regarding the conclusion; it should be in a separate paragraph or section demonstrating a more clear "clinical" application of the results found.

Reply:

We agree with the reviewer's opinion that a separate paragraph on clinical application would be important. Therefore, we have added an appropriate paragraph and inserted it on line 631-646.

  • There should be a section for the abbreviations.

Reply:

We understand that a list of abbreviations is very useful, since many abbreviations are included in the present manuscript. Unfortunately, the layout of the Cells-journal does not provide for this, but recommends an abbreviation at the first mention in the text.

  • No number for the reference in Line 588.

Reply:

The reference in Line 588 is number 53 and refers to the following article:  

Rashid, S.; Genest, J. Effect of obesity on high-density lipoprotein metabolism. Obesity (Silver Spring) 2007, 15, 2875–2888, doi:10.1038/oby.2007.342.

The number is included in the current reference list.

Reviewer 3 Report

This study is well-designed and the manuscript is well-written. I have no major issues to address regarding the methodology or results presented. I would only suggest the authors to provide more detailed data on the clinical impact and change in practice related to the authors findings.

Author Response

We thank all reviewers for their time to evaluate our manuscript and the useful advices for improvement! Please find our reply below.

Reviewer 3:

  • I would only suggest the authors to provide more detailed data on the clinical impact and change in practice related to the authors findings.

Reply:

This comment is identical to a comment made by reviewer 2. As noted above, we have added a paragraph in the discussion section that clearly describes the clinical relevance of the issue and FGF21 on line 631-646.

Reviewer 4 Report

In this study, Fritzi et al. reported the role of the Hedgehog pathway in obesity by regulating fibroblast growth factor 21. Overall, the study was well-designed and presented.  Some minor comments about the manuscript.

For diet composition, is kJ% or Kcal%?

Line 287, **** < 0.0001. > **** p < 0.0001.

In line 340, Check all the abbreviations, triacylglyceride (TAG) has been shown in line 182; type 2 diabetes showed several times across the manuscript; diacylglyceride 341 (DAG).

Line 506, (Fig. 1, 2) > (Figure 1, 2).

In addition, a brief discussion for the role of HGF21 and Hedgehod transcription factor GLI1 in human obese patients, Studies such as PMID: 33673326, only as an example.

Author Response

We thank all reviewers for their time to evaluate our manuscript and the useful advices for improvement! Please find our reply below.

Reviewer 4:

  • For diet composition, is kJ% or Kcal%?

Reply:

The indication of the dietary components is kJ%, as described on page 3 in Material and Methods. The resulting available energy of the diet is given in kJ per gram of the diet.

  • Line 287, **** < 0.0001. > **** p < 0.0001.

Reply:

We thank the reviewer for his hint. We have added the missing “p”.

  • In line 340, Check all the abbreviations, triacylglyceride (TAG) has been shown in line 182; type 2 diabetes showed several times across the manuscript; diacylglyceride 341 (DAG).

Reply:

We thank the reviewer for the good advice. We have consequently checked the text and implemented a uniform syntax for all lipid species and a consistent abbreviation at the first mention in the text.

  • Line 506, (Fig. 1, 2) > (Figure 1, 2).

Reply:

We thank the reviewer for his hint. We have added the missing letters.

  • In addition, a brief discussion for the role of HGF21 and Hedgehod transcription factor GLI1 in human obese patients, Studies such as PMID: 33673326, only as an example.

Reply:

Unfortunately, we are unable to link a reference to HGF21 and Hedgehog signalling pathway. Probably HGF21 is a typo. Based on the proposed paper, we assume that the reviewer most likely refers to Hhip1 (Hedgehog-interacting protein 1). We have expanded the discussion and included this and a second literature for this topic in the reference list (line 605-609).